# The First Step to Initiate Pediatric Palliative Care: Identify Patient Needs and Cooperation of Medical Staff

**DOI:** 10.3390/healthcare10010127

**Published:** 2022-01-09

**Authors:** Su Hyun Bae, Yeo Hyang Kim

**Affiliations:** 1Pediatric Palliative Care Center, Kyungpook National University Children’s Hospital, Daegu 41404, Korea; b_suhyun@naver.com; 2Department of Social Welfare, Kyungpook National University, Daegu 41566, Korea; 3Department of Pediatrics, School of Medicine, Kyungpook National University, Daegu 41566, Korea

**Keywords:** palliative care, children, life-limiting condition

## Abstract

Few Korean hospitals had experience in pediatric palliative care. Since the beginning of the national palliative care project, interest in pediatric palliative care has gradually increased, but the establishment of professional palliative care is still inadequate due to a lack of indicators. This study aimed to find considerations in the process of initiating palliative care services. The general and clinical characteristics of 181 patients aged less than 24 years who were registered at the pediatric palliative care center from January 2019 to August 2021 were evaluated. Life-limiting condition group 1 had the largest number of patients. The primary need for palliative care was psychological and emotional support, followed by information sharing and help in communication with the medical staff in decision-making processes. Seventy-two patients were technologically dependent, with one to four technical supports for each patient. The registration of patients with cancer increased with time, and the time from disease diagnosis to consultation for pediatric palliative care service was significantly reduced. In conclusion, before starting pediatric palliative care, it is necessary to understand the needs of patients and their families and to cooperate with medical staff.

## 1. Introduction

Advances in medical technology have increased the survival rate of pediatric and adolescent patients affected by life-threatening diseases or life-limiting conditions (LLC) [1]. However, due to complex disabilities, instead of achieving complete recovery, many patients with these diseases depend on medical devices or require intensive medical care [2,3]. In addition, patients with such diseases experience physical, psychological, social, and spiritual distress due to long-term intensive care. Despite the increasing survival rate, the quality of life of patients and their families is significantly reduced. The United States and European countries have established and provided palliative care services on a national scale to address this issue [4,5]. South Korea has also planned a palliative care system for children and adolescents, and in July 2018, two children’s hospitals in Seoul, South Korea, started pilot palliative care projects. As of 2021, nine hospitals in South Korea are participating in pilot projects (Figure 1).

In the United States, there are approximately 500,000 children and adolescents with LLC, and approximately 10% die each year [6]. In Japan, approximately 110,000 children and adolescents cope with LLC [7]. In Korea, there are approximately 130,000 children and adolescents aged less than 24 years with LLC [8]. Among them, 26.2% are patients with cancer, whereas the rest have noncancer-related diseases, such as congenital heart, genetic metabolic, neuromuscular, and degenerative diseases. While most adults who require palliative care are terminally ill, pediatric palliative care patients have various diseases.

Even if the same disease occurs, the demand for pediatric palliative care services may vary depending on the severity of the disease or stage of progression. While there is a high demand for pain control in patients with cancer, there may also be a high demand for control of neurological and respiratory symptoms in the nervous system and genetic metabolic diseases. Patients with neurological problems may require urgent physical care. Furthermore, psychosocial and spiritual care of patients and their families is essential [9]. Therefore, due to the need and specificity of pediatric palliative care, there is a limit to providing a standardized palliative care program. The guidelines for Korean pediatric palliative care projects provide recommendations for the provision of individually customized care services to patients.

Few Korean hospitals were experienced in pediatric palliative care. Except for Seoul, no other city offered pediatric palliative care services. Since the beginning of the national project, interest in pediatric palliative care has gradually increased, but the establishment of professional palliative care is still inadequate due to a lack of indicators. Therefore, it is necessary to share experiences in establishing palliative care services suitable for the region for a short period in children’s hospitals that lack experience in pediatric palliative care.

This study aimed to find factors to be considered in the process of initiating palliative care services by analyzing the characteristics of patients registered at the first pediatric palliative care center established outside Seoul.

## 2. Materials and Methods

### 2.1. Participants

This study enrolled 181 patients aged less than 24 years who were registered at the Pediatric Palliative Care Center at Kyungpook National University Children’s Hospital (Daegu, Korea) from January 2019 to August 2021.

The general and clinical characteristics of the patients were retrospectively reviewed from medical records, basic registration information, initial counseling records, medical progress, and withdrawal records. Patient information was classified into three categories: (1) general characteristics, such as sex, age, residence, and insurance type; (2) clinical characteristics, such as final diagnosis, date of diagnosis, and LLC classification; and (3) palliative care-related information, such as consultation date, the reason for consultation, and medical requirement.

To detect changes in the behavior of medical staff regarding pediatric palliative care consultation, the time taken from diagnosis to palliative care consultation and the number of newly registered patients with cancer and those with noncancer diseases were compared. The period was assessed by dividing it into two periods of 6 months each: 6 months from the start of the local pediatric palliative care center (January–June 2019, period 1) and the last 6 months after 3 years (January–June 2021, period 2).

### 2.2. Definition

In this study, LLC were classified into four categories based on the disease characteristics [10,11] as follows: group 1, life-threatening conditions for which curative treatment is feasible but can fail; group 2, conditions for which premature death is inevitable; group 3, progressive conditions without curative treatment options; and group 4, irreversible but nonprogressive conditions causing severe disability, with a high risk of premature death from an unpredictable life-threatening event or episode.

The reasons for the need for palliative care were classified into six categories as follows: (1) symptom and pain management, (2) support for decision-making processes, (3) psychological and emotional support for patients and their families, (4) socioeconomic support, (5) nursing and home care adjustment, and (6) end-of-life decision and care.

The technical supports for patients with medical complexity were classified into home ventilator application, home oxygen therapy, tracheal suction, home parenteral nutrition, and enteral nutrition through nasogastric or gastrostomy tube.

### 2.3. Statistical Analyses

All statistical analyses were conducted using IBM SPSS Statistics for Windows, version 26.0 (IBM Co., Armonk, NY, USA). The independent t-test and chi-square tests were used to evaluate differences between patients with cancer and noncancer diseases and between periods. A *p*-value of <0.05 was considered statistically significant.

## 3. Results

### 3.1. Patients’ General Characteristics

Table 1 shows the general and clinical characteristics of the 181 patients registered at the pediatric palliative care center.

The patients included 100 males (55%) and 81 females (45%). The most common age group was from 1 to 5 years old. Diseases were classified into cancer and noncancer, with more patients having noncancer diseases. Eighty patients were diagnosed before the pediatric palliative care service began in January 2019, and 101 patients were diagnosed after that.

Most patients lived in the main city (Daegu), where pediatric palliative care centers are located, and in nearby areas (Gyeongsangbuk-do) (Figure 2), and 33% of patients were linked to hospitals in Seoul. All patients were supported by National Health Insurance. Among the LLC groups, groups 1 and 2 had the largest and smallest number of patients, respectively.

### 3.2. Primary Reasons for the Need for Palliative Care 

Table 2 shows the reasons for the need for palliative care.

The reasons for the need for palliative care overlapped, and the highest priority was given to psychological and emotional support, followed by information sharing and communication with the medical staff in decision-making processes, such as future treatment plans. Although the National Health Insurance supported all patients, 24% of patients required additional economic support. Furthermore, approximately 10% of patients wanted to make decisions related to life-sustaining treatments, nursing homes, and end-of-life care. 

These reasons for the need for palliative care did not indicate a significant difference between patients with cancer and those with noncancer diseases.

### 3.3. Technical Support for Patients with Medical Complexity

Technical support was required for patients with medical complexity registered at the pediatric palliative care center. Seventy-two patients had technology dependency, of which only one was cancer patient. Technical support overlapped, with one to four of the five technical supports for each patient (Figure 3).

Enteral nutrition through a nasogastric or gastrostomy tube was the most common technical support, at 89%, followed by respiratory support with simple or high flow nasal cannula at 74% and home ventilator at 56%.

### 3.4. Changes in the Behavior of Medical Staff Regarding Pediatric Palliative Care

Compared with period 1, period 2 did not show a significant difference in the number of registered patients (period 1, 32; period 2, 31). Although the proportion of patients with cancer was not significantly different between the periods (period 1, 9; period 2, 15, *p* = 0.098), the number of registered patients with cancer increased in period 2.

There were 18 newly diagnosed patients in period 1 and 20 in period 2. The time from the first disease diagnosis to consultation for pediatric palliative care service was significantly reduced from 5.2 ± 6.8 days in period 1 to 1.9 ± 1.4 days in period 2 (*p* = 0.036).

## 4. Discussion

This study showed that although Korea’s pediatric palliative care service is still in its developing stages, the understanding of the characteristics of pediatric diseases that need palliative care and the medical staff’s perception that palliative care may be considered alongside treatment is rapidly advancing.

In the 1960s, the primary focus of pediatric palliative care was within the hospice to enhance the quality of death [12]. Because hospice care is primarily intended for adult patients with terminal cancer, pediatric palliative care also has a strong perception of death. However, recently, hospice care has transitioned to palliative care, and palliative care services focus on “living” and not “dying” [5,13,14]. In other words, palliative care focuses on improving the quality of life in addition to medical care that includes a cure for patients with severe illnesses. In this study, only 5% of patients required hospice care services, and the primary reasons for the need for palliative care were psychological and emotional support, decision-making process service, and socioeconomic support.

It is important to provide psychoemotional support to patients and their families receiving pediatric palliative care [9,15,16]. The diagnosis of LLC is a significant crisis for patients and their families. Patients with LLC who have to endure pain can experience psychological difficulties such as depression and anxiety. Parents who take care of their sick children continue their daily lives tending to their children, experiencing complex emotions such as anxiety, guilt, and anger due to their children’s illness; thus, they become physically and psychologically exhausted. Therefore, it is vital to manage symptoms and provide psychological support together with pediatric palliative care. This suggests that pediatric palliative care should be considered differently from adult palliative care.

Children and adolescents who require palliative care are affected by numerous diseases, so palliative care services suitable for each case profile should be made available [2,17,18]. In this study, the incidence of cancer and noncancer diseases in pediatric palliative care registered patients was 41% and 59%, respectively, similar to the incidences of 45% and 55%, respectively, reported by the National Hospice Center. Since Korean hospice palliative care services began with care for adult patients with terminal cancer, and the scope of service expanded to AIDS, cirrhosis, and chronic obstructive pulmonary disease, but there is still a lack of awareness of the need for palliative care for patients with noncancer diseases. Pediatric patients have a higher rate of noncancer diseases, such as congenital, genetic metabolic, neuromuscular, and degenerative diseases, efforts are needed to provide customized palliative care services reflecting the needs of patients and their families.

Especially, patients with noncancer diseases were found to have high technological dependence and required home care [2,19,20]. They need active care due to breathing or nutritional problems, and there are many medical demands related to technological dependence. In Korea, pediatric home care services for severely ill children started in January 2019 so that patients would be stable at home following discharge. Home care services can effectively manage pediatric and adolescent palliative care patients by educating their families on how to care for their children and periodically checking on the patient’s condition, such as through phone counselling or medical staff visits to homes if necessary. This study was conducted at the pediatric palliative care center that was capable of providing home care services, and patients who need these services often wanted palliative care services.

Pediatric patients with oncological diseases were not discussed or implemented until the time of death approached despite efforts for early initiation of palliative care [21,22]. A retrospective systematic review found that from 1999 to 2016, 54.5% of pediatric patients with oncological diseases received palliative care immediately before death [23]. This study showed that as experience and understanding about pediatric palliative care increased, the number of patients receiving palliative care concurrently with a diagnosis also increased. Although the total number of registered patients did not differ among the periods, the number of patients with cancer registered in the last 6 months showed an increasing trend compared with that in the first 6 months of palliative care service. In particular, the significant reduction in the period from primary diagnosis to palliative care counseling is thought to be related to a change in the medical staff’s perception that the need for palliative care needs to be combined with diagnosis and treatment.

In Korea, adult palliative care was initially started as a clinic in 1965; currently, 109 institutions provide adult palliative care. Pediatric palliative care started in 2018, and nine institutions have opened pediatric palliative care centers until 2021 [24]. Although the supply is insufficient compared with the demand for pediatric palliative care, the field has developed rapidly 3 years since the pediatric palliative care pilot project began. The Korean National Hospice Center reported that the quality of life of 781 children and adolescent patients with LLC and their families has improved with palliative care [24]. 

To develop pediatric palliative care services, it is necessary to improve access to services to meet the needs of patients and their families. The first local pediatric palliative care center is located in the third-largest city in Korea. Following a review of the distribution of registered patient residences, the first local pediatric palliative care center was in charge of patients residing in nearby areas and makes it easier for local patients to access services. In addition, clinical studies that conduct surveys or interviews on patients with LLC and their family needs, professionals training through customized education, and medical staff perceptions of pediatric palliative care are needed.

The limitations of this study include the fact that it was a retrospective review of medical records and that it was difficult to observe a consistent pattern of change due to the limited quantitative growth of pediatric palliative care centers as a result of the COVID-19 pandemic. It is difficult to generalize the results as this study was conducted at a single local pediatric palliative care center. Other methods such as questionnaires to investigate changes in the behavior of the medical staff were not employed. Furthermore, human and hospital internal and external factors that influenced this study were not considered. 

## 5. Conclusions

In order for hospitals with no experience in pediatric palliative care to provide pediatric palliative care, knowledge and information related to regional background, needs of patients and families, patient technical dependence, and understanding of palliative care by medical staff are required. These will help new centers provide the most functional pediatric palliative care services as soon as possible.

## Figures and Tables

**Figure 1 healthcare-10-00127-f001:**
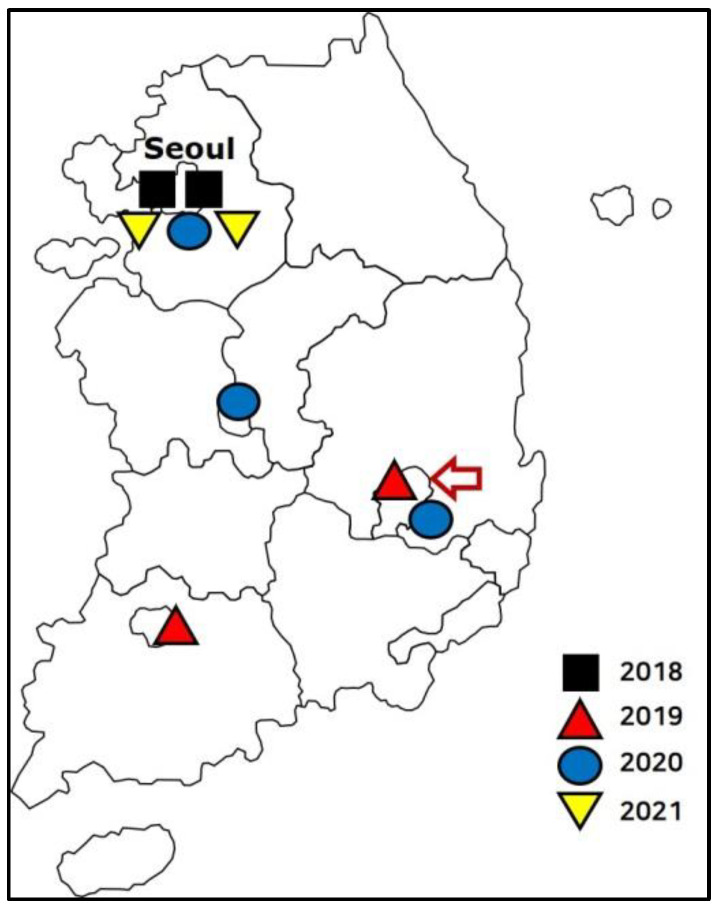
Pediatric palliative care centers in South Korea. Centers that opened in 2018 (
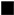
), 2019 (
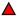
), 2020 (
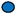
), and 2021 (
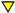
) are shown. The red arrow indicates the first local pediatric palliative care center.

**Figure 2 healthcare-10-00127-f002:**
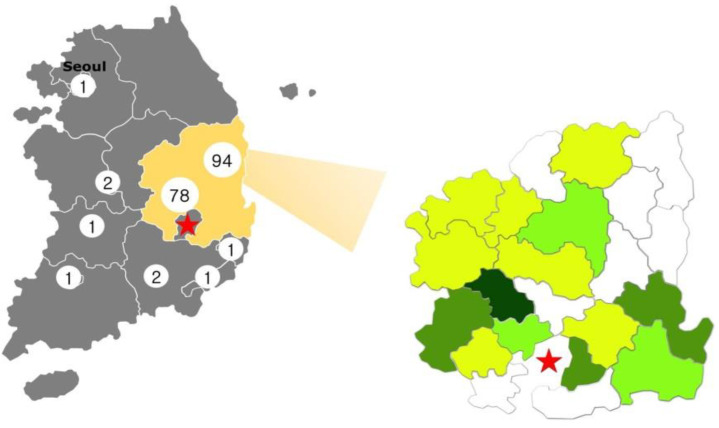
Patient residential areas and the number of patients. The red star indicates the main city, Daegu, with the first local pediatric palliative care center. The yellow zone indicates nearby areas (Gyeongsangbuk-do). Color severity is associated with the number of patients. Dark-green implies > 20 patients, green > 10, light-green > 5, and yellowish-green < 5. The gray zone indicates other areas where patients reside.

**Figure 3 healthcare-10-00127-f003:**
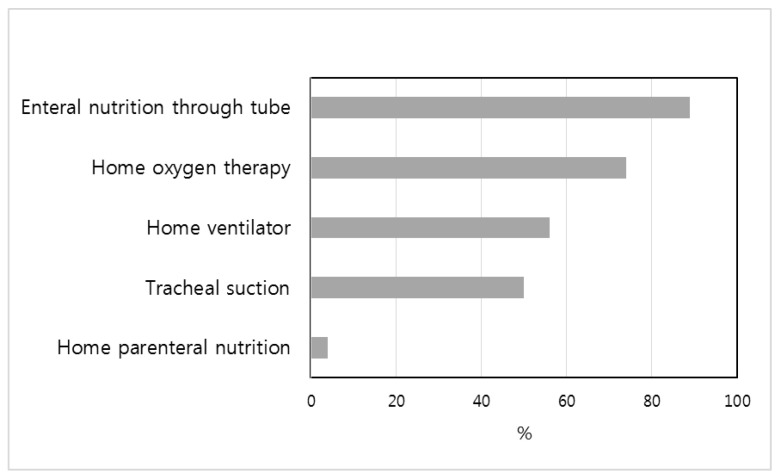
Technical support for patients with medical complexity.

**Table 1 healthcare-10-00127-t001:** General and clinical characteristics of the patients.

Characteristic	Number (%)
Sex	
Male	100 (55)
Female	81 (45)
Age (years)	
<1	10 (5)
1–5	64 (35)
6–12	54 (30)
13–18	43 (24)
19–24	10 (6)
Type of diagnosis	
Cancer	75 (41)
Noncancer	106 (59)
City of residence	
Daegu	78 (43)
Gyeongsangbuk-do	94 (52)
Others	9 (5)
Link with hospitals in the capital	59 (33)
LLC group	
group 1	111 (62)
group 2	11 (6)
group 3	24 (13)
group 4	35 (19)

Abbreviation: LLC, life-limiting condition; group 1, life-threatening conditions for which curative treatment is feasible but may fail; group 2, conditions in which premature death is inevitable; group 3, progressive conditions without curative treatment options; group 4, irreversible but nonprogressive conditions causing severe disability, with a high risk of premature death from an unpredictable fatal event or episode.

**Table 2 healthcare-10-00127-t002:** Primary reasons for the need for palliative care.

Category	Cancer Number (%)	Noncancer Number (%)	*p*-Value
Psychological and emotional support	71 (39)	93 (51)	0.77
Decision-making process service	13 (7)	52 (29)	0.97
Socioeconomic support	9 (5)	34 (19)	0.91
Symptom and pain management	14 (8)	22 (12)	0.82
Life-sustaining treatment decision service	9 (5)	10 (6)	0.79
Nursing and home care service	2 (1)	7 (4)	0.83
Hospice care service	5 (3)	4 (2)	0.80

## Data Availability

The data presented in this study are available on request from the corresponding author.

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
