# Peer review of "The First Step to Initiate Pediatric Palliative Care: Identify Patient Needs and Cooperation of Medical Staff"

_healthcare, 2022, doi:10.3390/healthcare10010127_

Round 1
Reviewer 1 Report
The authors said that the aim of this study was to find factors that should be considered when starting pediatric palliative care, but the results and discussions did not cover this sufficiently. Also, how to successfully set up palliative care as mentioned in the title is not included in the manuscript. Finally, this manuscript requires extensive editing of the English language.
Author Response
The authors said that the aim of this study was to find factors that should be considered when starting pediatric palliative care, but the results and discussions did not cover this sufficiently. Also, how to successfully set up palliative care as mentioned in the title is not included in the manuscript. Finally, this manuscript requires extensive editing of the English language.
>> Before submission, the manuscript has been carefully reviewed by an experienced editor “https://www.enago.co.kr” whose first language is English and who specializes in editing papers written by scientists whose native language is not English. Enago re-edited the manuscript, and the authors have attached an English language editing certificate.
Reviewer 2 Report
Overall feedback
The effort invested in this study is appreciated. This study was conducted to inform the current status of pediatric palliative care provision at the national level in South Korea and to determine factors to be considered for effectively promoting pediatric palliative care. The results of the data record analysis in this study suggest the importance of understanding patients’ needs and changing the medical staff’s perception in order to successfully initiate pediatric palliative care. The following points need to be considered to improve the paper.
<Introduction>
- Please describe and compare the level of burden of disease, such as the prevalence of pediatric life-limiting conditions (LLC) in South Korea and overseas and the mortality rates in these patients.
<Method>
- What are the criteria for dividing period 1 and period 2?
- What are the survey indicators that measure changes in the behavior of medical staff? In the Materials and Methods section, please provide the research methodology to measure changes in staff behavior.
- Please add details on the IRB approval for the study and the acquisition of participant informed consent.
<Results>
- The study shows that the independent t-test and chi-square test were used; however, the results of the hypothesis test are not presented in the tables and figures.
- Looking at the distribution of data in Figure 3, one would expect to see a significant difference between cancer patients and non-cancer patients; therefore, please provide the p-value.
<Discussion>
- Please add the strengths and limitations of this study. For example, are there any limitations regarding generalizability of the results for a single hospital located outside the metropolitan area?
- Please consider the possible research avenues and tasks for a follow-up of this study.
- Please described what should be emphasized in the field of pediatric palliative care in south Korea, an advanced country, three years after the establishment of a national pediatric palliative care system.
Author Response
Thank you for the review of our manuscript healthcare-1506141, titled “Ways to Successfully Initiate Pediatric Palliative Care: Identify Patient Needs and Change the Perception of Medical Staff.” We have made corrections/modifications to our manuscript in response to the comments raised by the Editor and the reviewers. Point-by-point responses to these comments are indicated hereafter in blue. Changes are yellow-highlighted in the revised manuscript.
*******************************************************************************************************************
Reviewer #2
The effort invested in this study is appreciated. This study was conducted to inform the current status of pediatric palliative care provision at the national level in South Korea and to determine factors to be considered for effectively promoting pediatric palliative care. The results of the data record analysis in this study suggest the importance of understanding patients’ needs and changing the medical staff’s perception in order to successfully initiate pediatric palliative care. The following points need to be considered to improve the paper.
<Introduction>
- Please describe and compare the level of burden of disease, such as the prevalence of pediatric life-limiting conditions (LLC) in South Korea and overseas and the mortality rates in these patients.
>> Thank you very much for the reviewer’s comment. We have mentioned the number of patients with LLC and deaths in South Korea and overseas with additional references as follows: “In the United States, there are approximately 500,000 children and adolescents with LLC, and approximately 10% die each year [6]. In Japan, approximately 110,000 children and adolescents cope with LLC [7].” (Lines 39–41)
<Method>
- What are the criteria for dividing period 1 and period 2?
>> We apologize for not explaining the criteria for each period. We have added a description of the definition of periods 1 and 2 as follows: “The period was assessed by dividing it into two periods of 6 months each: 6 months from the start of the local pediatric palliative care center (January–June 2019, period 1) and the last 6 months after 3 years (January–June 2021, period 2).” (Lines 86–89)
- What are the survey indicators that measure changes in the behavior of medical staff? In the Materials and Methods section, please provide the research methodology to measure changes in staff behavior.
>> Thank you for the reviewer’s comment. We have described the research methodology to measure changes in staff behavior as follows: “To detect changes in the behavior of medical staff regarding pediatric palliative care consultation, we compared the period from diagnosis to palliative care consultation and the ratio of patients with cancer and those with noncancer diseases.” (Lines 84–86)
- Please add details on the IRB approval for the study and the acquisition of participant informed consent.
>> We retrospectively reviewed the general and clinical characteristics of the patients through medical records, and medical record analysis did not require participant consent. Further, when a patient is enrolled in a palliative care center, we receive a consent form (Personal Information Consent form) provided by the government in advance.
<Results>
- The study shows that the independent t-test and chi-square test were used; however, the results of the hypothesis test are not presented in the tables and figures.
>> We apologize for not explaining the results of the hypothesis test in detail. We have changed Figure 3 to Table 2 in “3.2 Primary reasons for the need for palliative care” and have presented the results of the independent t-test. (Line 149)
- Looking at the distribution of data in Figure 3, one would expect to see a significant difference between cancer patients and non-cancer patients; therefore, please provide the p-value.
>> First, Figure 3 was changed to Table 2 in “3.2 Primary reasons for the need for palliative care.” There was no significant difference between patients with cancer and without cancer. We have added the p-value. (Line 142).
<Discussion>
- Please add the strengths and limitations of this study. For example, are there any limitations regarding generalizability of the results for a single hospital located outside the metropolitan area?
>> Thank you very much for the reviewer’s comment. Accordingly, as a limitation of this study, we have added the following sentence: “It is difficult to generalize the results as this study was conducted at a single local pediat-ric palliative care center.” (Lines 246–247)
- Please consider the possible research avenues and tasks for a follow-up of this study.
>> Thank you for pointing this out. We considered and provided suggestions for follow-up research as follows: “Future clinical studies that conduct surveys or interviews on patients with LLCs and their family needs as well as medical staff perceptions of pediatric palliative care are warrant-ed.” (Lines 250–252)
- Please described what should be emphasized in the field of pediatric palliative care in south Korea, an advanced country, three years after the establishment of a national pediatric palliative care system.
>> We appreciate and agree with the reviewer’s point. We have mentioned the points to be emphasized in the field of pediatric palliative care in South Korea as follows: “In Korea, adult palliative care was initially started as a clinic in 1965; currently, 109 institutions provide adult palliative care. Pediatric palliative care started in 2018, and nine institutions have opened pediatric palliative care centers until 2021 [24]. Although the supply is insufficient compared with the demand for pediatric palliative care, the field has developed rapidly 3 years since the pediatric palliative care pilot project began. The Kore-an National Hospice Center reported that the quality of life of 781 children and adolescent patients with LLC and their families have improved with palliative care [24]. To develop pediatric palliative care services, it is necessary to improve access to services to meet the needs of patients and their families. In addition, professionals should be trained through customized education.” (Lines 233–242)
*******************************************************************************************************************
We wish to extend our sincere thanks to the Editor and reviewers for the helpful comments and the time dedicated to reviewing our manuscript. We hope that this revision will satisfy the comments and requests from the Editor and reviewers as well as improve the overall quality of our manuscript. We also hope that the revised manuscript is now acceptable for publication in Healthcare.
Reviewer 3 Report
Line 3: ¿How do you evaluate the perception of staff? If you don`t evaluate this matter, title should change.
Line 40: How do you get this number? Do you have a registry of this patients? It is an estimation?
Line 78: There is not a unique reason to refer patients. It could be an unexactly conclusion.
Line 124: In table 1 there is a mistake in a sign: 19< should be change to 19>
Clasifficating the pediatric patients in two groups (cancer and not cancer) is not a good practice. It becomes from adult palliative focusing. I understand it at the beginning of pediatric palliative services to compare with adult services.
Line 168 and 171: "Our institution": In scientific texts is preferable not to use first person
Line 179: We dont know how many patientes died in order to evaluate if 10% is adequate.
Line 210: "Our center": In scientific texts is preferable not to use first person
Line 224: It is reasonble to say this in the discussion but is too much for the title.
Author Response
Thank you for the review of our manuscript healthcare-1506141, titled “Ways to Successfully Initiate Pediatric Palliative Care: Identify Patient Needs and Change the Perception of Medical Staff.” We have made corrections/modifications in our manuscript in response to the comments raised by the Editor and reviewers. Point-by-point responses to these comments are indicated hereafter in blue. Changes are yellow-highlighted in the revised manuscript.
*******************************************************************************************************************
Reviewer #3
Line 3: How do you evaluate the perception of staff? If you don`t evaluate this matter, title should change.
>> Thank you for pointing this out. We have revised the title by correcting “Perception” to “Cooperation.” (Line 3)
Line 40: How do you get this number? Do you have a registry of this patients? It is an estimation?
>> We apologize for the unclear source. We have added a new reference, reference 8. (Line 42)
Line 78: There is not a unique reason to refer patients. It could be an unexactly conclusion.
>> Thank you for the reviewer’s comment. The expression “re-referral rate to local pediatric palliative care centers” means “transfer of patient from one palliative care center to another palliative care center due to the patient’s main residency.” However, we did not acquire accurate data and excluded this from the analysis. We deleted the phrase “re-referral rate to local pediatric palliative care centers.”
Line 124: In table 1 there is a mistake in a sign: 19< should be change to 19>
>> This has been corrected to “19–24” because it refers to children and adolescent patients aged less than 24 years. (Line 132)
Clasifficating the pediatric patients in two groups (cancer and not cancer) is not a good practice. It becomes from adult palliative focusing. I understand it at the beginning of pediatric palliative services to compare with adult services.
>> Thank you very much for the reviewer’s comment. Patients with LLC are sometimes divided into four groups, but many previous studies classified pediatric palliative care patients into two groups (cancer/non-cancer) in South Korea and overseas. Currently, there is a lack of awareness of pediatric palliative care compared with adult hospice, and we used two groups (cancer/noncancer) for convenience and to improve reader comprehension.
Line 168 and 171: "Our institution": In scientific texts is preferable not to use first person.
>> Thank you for pointing this out. We have corrected this to “First local pediatric palliative care center.” (Lines 175 and 178)
Line 179: We don’t know how many patients died in order to evaluate if 10% is adequate.
>> First, we apologize for not explaining the results in detail. We changed Figure 3 to Table 2 in “3.2 Primary reasons for the need for palliative care” and presented the results of the independent t-test. (Line 149) According to Table 2, the sentences in Line 179 were revised as follows” “In this study, only 5% of patients required hospice care services, and the primary reasons for the need for palliative care were psychological and emotional support, deci-sion-making process service, and socioeconomic support.” (Lines 186–188)
Line 210: "Our center": In scientific texts is preferable not to use first person.’
>> Thank you for pointing this out. We have corrected this to “This study was conducted at the pediatric palliative care center.” (Line 218)
Line 224: It is reasonable to say this in the discussion but is too much for the title.
>> Thank you very much for the reviewer’s comment. We have revised the title by correcting “Perception” to “Cooperation.” (Line 3)
*******************************************************************************************************************
We wish to extend our sincere thanks to the Editor and reviewers for the helpful comments and the time dedicated in reviewing our manuscript. We hope that this revision will satisfy the comments and requests from the Editor and reviewers as well as improve the overall quality of this manuscript. We also hope that the revised manuscript is now acceptable for publication in Healthcare.